# Performance of a Differentiation of Infected from Vaccinated Animals (DIVA) Classical Swine Fever Virus (CSFV) Serum and Oral Fluid Erns Antibody AlphaLISA Assay

**DOI:** 10.3390/ani13243802

**Published:** 2023-12-09

**Authors:** Yaowalak Panyasing, Luis Gimenez-Lirola, Roongroje Thanawongnuwech, Phakawan Prakobsuk, Yanee Kawilaphan, Apisit Kittawornrat, Ting-Yu Cheng, Jeffrey Zimmerman

**Affiliations:** 1Department of Pathology, Faculty of Veterinary Science, Chulalongkorn University, Bangkok 10330, Thailand; roongroje.t@chula.ac.th; 2Animal Virome and Diagnostic Development Research Unit, Faculty of Veterinary Science, Chulalongkorn University, Bangkok 10330, Thailand; 3Department of Veterinary Diagnostic and Production Animal Medicine, College of Veterinary Medicine, Iowa State University, Ames, IA 50011, USA; luisggl@iastate.edu (L.G.-L.); jjzimm@isu.edu (J.Z.); 4CPF (Thailand) Public Company Limited, Bangkok 10120, Thailand; phakawan.pra@cpf.co.th (P.P.); yanee.kaw@cpf.co.th (Y.K.); apisit.k@cpf.co.th (A.K.); 5Department of Veterinary Preventive Medicine, College of Veterinary Medicine, The Ohio State University, Columbus, OH 43210, USA; cheng.1784@osu.edu

**Keywords:** AlphaLISA, classical swine fever virus, DIVA, Erns, IgG, oral fluid, serum

## Abstract

**Simple Summary:**

Classical swine fever (CSF) is a highly contagious viral disease causing significant economic losses to swine producers in most parts of the world. Previously, pseudorabies virus (PRV) was eliminated from commercial swine herds using gene-deleted vaccines and assays capable of differentiation of infected from vaccinated animals (DIVA). Modeling on this experience, we developed an assay able to differentiate CSFV-infected pigs from pigs vaccinated with a modified live CSFV marker vaccine (Suvaxyn^®^ CSF Marker). In brief, the CSFV Erns IgG AlphaLISA^®^ detects specific antibodies against a CSFV envelope glycoprotein (Erns) present in wild-type viruses. In contrast, the Erns glycoprotein is absent in the CSFV marker vaccine; hence, vaccinated pigs are negative on the CSFV Erns AlphaLISA^®^. Importantly for ease of use in the field and in the laboratory, the assay is able to test both serum and oral fluid samples. The combined use of a marker vaccine to protect against clinical disease and a serum/oral fluids DIVA assay able to identify wild-type CSFV infections will greatly improve the capacity of the pork industry to deal quickly and decisively with CSFV.

**Abstract:**

Classical swine fever virus (CSFV) is an OIE-listed disease that requires effective surveillance tools for its detection and control. The aim of this study was to develop and evaluate the diagnostic performance of a novel CSFV Erns IgG AlphaLISA for both serum and oral fluid specimens that would likewise be compatible with the use of CSFV E2 DIVA vaccines. Test performance was evaluated using a panel of well-characterized serum (n = 760) and individual (n = 528) or pen-based (n = 30) oral fluid samples from four groups of animals: (1) negative controls (n = 60 pigs); (2) inoculated with ALD strain wild-type CSFV (n = 30 pigs); (3) vaccinated with LOM strain live CSFV vaccine (n = 30 pigs); and (4) vaccinated with live CSFV marker vaccine on commercial farms (n = 120 pigs). At a cutoff of S/P ≥ 0.7, the aggregate estimated diagnostic sensitivities and specificities of the assay were, respectively, 97.4% (95% CI 95.9%, 98.3%) and 100% for serum and 95.4% (95% CI 92.9%, 97.0%) and 100% for oral fluid. The Erns IgG antibody AlphaLISA combined DIVA capability with solid diagnostic performance, rapid turnaround, ease of use, and compatibility with both serum and oral fluid specimens.

## 1. Introduction

Classical swine fever virus (CSFV) is a member of the genus *Pestivirus* and the family *Flaviviridae* [1]. Pestiviruses infectious for pigs include bovine viral diarrhea viruses 1 and 2, border disease virus, Bungowannah virus, and atypical porcine pestivirus (APPV or Pestivirus K), but classical swine fever (CSF) is the only OIE-listed disease in the group [2]. Although its viral etiology has been recognized since 1903 [3], CSFV remains endemic in many parts of the world. The CSFV imposes a significant economic impact on pig health and international trade, and periodically emerges in CSFV-free areas, e.g., the Netherlands in 1997 [4], Great Britain in 2000 [5], South Africa in 2005 [6], Israel in 2009 [7], and Japan in 2018 [8].

Differentiation of infected from vaccinated animals (DIVA)-enabled vaccines and companion antibody assays were pioneered for the control and/or elimination of Aujeszky’s disease (pseudorabies) virus and have been highly effective in this role [9,10,11]. Continuing on this path, research has been focused on the development of CSFV DIVA vaccines and differential tests [12]. Biologically compatible with this aim, CSFV’s positive-sense, single-stranded RNA genome contains a single open reading frame encoding a polyprotein (3898 amino acids) that undergoes co- and post-translation processing to produce four structural proteins (C, Erns, E1, E2) and seven non-structural proteins (Npro, p7, NS2-3, NS4A, NS4B, NS5A, NS5B) [13,14,15,16]. Protective immunity against CSFV relies on the production of neutralizing antibodies against structural proteins E2 and Erns [14]. This fact has been used in the development of DIVA CSFV E2 vaccines and their companion Erns-based ELISAs. That is, animals vaccinated with E2 vaccines produce antibodies against CSFV E2 and not Erns, while field virus-infected animals produce antibodies against both E2 and Erns.

CSFV Erns-based ELISAs have been developed for their use with CSF E2-based DIVA vaccines [17], but studies of commercially available DIVA CSFV ELISAs underline the need for further improvement in their diagnostic sensitivity and specificity [18,19,20,21]. Alternatively, the bead-based AlphaLISA (amplified luminescent proximity homogenous assay) antibody format reportedly achieves both high analytical sensitivity and rapid turnaround because of short incubation times and the lack of wash steps [22]. The use of powerful analytical platforms such as AlphaLISA, coupled to specific target antigens and DIVA vaccines, allows for the development of highly specific and sensitive DIVA tests, which are also user-friendly and reproducible. AlphaLISAs have previously been developed for the detection of antibodies against the porcine epidemic diarrhea virus [23] and African swine fever virus [24]. The objective of the present study was to develop a CSFV Erns IgG AlphaLISA and evaluate its performance using serum and oral fluid samples from pigs of known CSFV antibody status.

## 2. Materials and Methods

### 2.1. Experimental Design

Specimens used for the analysis of the CSFV Erns IgG AlphaLISA included serum and oral fluid samples from pigs inoculated with wild-type (ALD strain) or live vaccine (LOM strain) CSF viruses in research facilities at Chulalongkorn University (Bangkok, Thailand; IACUC #1473003). In addition, to further evaluate the test’s DIVA capabilities, field samples (serum and oral fluid) from commercial pigs vaccinated with a live recombinant CSFV marker live attenuated vaccine (Suvaxyn^®^ CSF Marker, Zoetis Belgium SA, Louvain-la-Neuve, Belgium) were also included in this study. The CSFV marker vaccine is based on an infectious cDNA clone of BVDV strain CP7 in which the BVDV E2 was exchanged with the E2 from CSFV Alfort/187 [17]. The presence of CSFV antibody in samples from inoculated or vaccinated pigs was confirmed by the serum virus neutralization test (VNT). Finally, the performance of the CSFV AlphaLISA was determined by receiver operating characteristic (ROC) analyses of the serum and oral fluid testing data.

### 2.2. Biological Samples

A total of 1318 well-characterized serum (n = 760) and oral fluid (n = 558) samples were used in the study: Group 1. CSFV-naïve pigs (serum n = 160, individual pig oral fluid n = 160); Group 2. CSFV-naïve pigs inoculated with CSFV ALD wild-type strain (serum n = 100, individual pig oral fluid n = 184); Group 3. CSFV-naïve pigs vaccinated with live CSFV LOM strain vaccine (serum n = 100, individual pig oral fluid n = 184); Group 4. CSFV-naïve pigs vaccinated with Suvaxyn^®^ CSFV marker vaccine (serum n = 360, pen-based oral fluids n = 30).

Groups 1, 2, and 3 samples were collected from animals housed in research facilities at Chulalongkorn University using procedures and methods described elsewhere [25]. In brief, Group 1 samples were collected prior to vaccination or inoculation of Groups 2 and 3. Groups 2 (n = 30 pigs) and 3 (n = 30 pigs) samples were collected from pigs inoculated with wild-type (ALD strain; 10^5^ TCID_50_/mL) CSFV or vaccinated with live vaccine virus (LOM strain; 10^3^ TCID_50_/mL), respectively. The serum and oral fluid samples used in the study were collected between days post-inoculation (DPIs) or days post-vaccination (DPVs) 14 and 28 [25]. Oral fluid samples were collected as described elsewhere [25]. Group 4 samples were collected from pigs in commercial swine production systems intramuscularly vaccinated with the Suvaxyn^®^ CSF Marker vaccine (CP7_E2alf; 10^4.8^ to 10^6.5^ TCID_50_/mL) at 9 weeks of age and sampled at DPVs 14, 30, and 60. At each collection, serum was collected from 120 pigs, and pen-based oral fluids were collected from 10 pens, each holding 50 to 60 pigs.

### 2.3. Virus Neutralization Test (VNT)

Virus neutralization testing was performed as previously described [25]. Briefly, the test was performed in 96-well microtiter plates. Minimum essential medium (MEM; 50 µL; Life Technologies, Paisley, UK) was added to all wells. Then, serum (50 µL) was added and serially 2-fold diluted. Thereafter, stock CSFV strain ALD (50 µL; 300 TCID_50_/50 µL) was added to each well and the plate incubated for 1 h (37 °C, 5% CO_2_). Then, SK6 cells (100 µL; 3 × 10^5^ cells/mL) were added to each well and the plate incubated for 72 h (37 °C, 5% CO_2_). Subsequently, the plate was tested with neutralizing peroxidase-liked assay (NPLA) as previously described [25] to determine the neutralizing antibody titer. Neutralizing antibody titers were measured as the reciprocal of the highest dilution of serum that completely inhibited viral infection, with neutralizing antibody titers ≥ 2 (log_2_ 1) considered positive for CSFV antibody.

### 2.4. CSFV Erns Antigen

Erns antigen was produced as previously described [26]. In brief, the codified region of the truncated N-terminal CSFV Erns gene was synthetically produced (Shanghai Genery Biotech Co., Ltd., Shanghai, China) with the addition of a 5′ GP67 terminal signal peptide for expression (Bac-to-Bac™ Baculovirus Vector System, Invitrogen, Carlsbad, CA, USA), followed by a hexa-histidine tag and a “GS linker” for protein purification by affinity chromatography. The PCR-amplified Erns region was cloned into pFastBac1 (Invitrogen) using EcoRI and XhoI restriction enzymes, and confirmed by digestion and sequencing. CSFV Erns protein was purified from culture supernatant by Ni-chelating SFF affinity chromatography (GE Healthcare), according to the manufacturer’s instructions. Protein elution was dialyzed and analyzed by SDS-PAGE and Western blot. Erns molecular weight measured by SDS-PAGE was 25–35 KDa. The final concentration of Erns protein was 0.75 mg/mL.

### 2.5. CSFV Erns AlphaLISA Bead Conjugation

The protein buffer was exchanged with PBS (0.1 M NaKPO_4_ pH 8.0) on a Zeba^TM^ spin column (Thermo Fisher Scientific, Rockford, IL, USA) and concentrated up to 1 mg/mL protein (BCA protein assay). AlphaLISA acceptor beads were washed with the protein buffer (pH 8) following the manufacturer’s protocol. The Erns protein was conjugated to raw acceptor beads via reductive amination, as described by the manufacturer (PerkinElmer Health Sciences, Inc., Boston, MA, USA). Briefly, 0.1 mg of Erns protein (1 mg/mL, 100 µL) was mixed with 1 mg AlphaLISA^®^ beads and reaction buffer (100 mM HEPES pH 7.4) in a 1.5 mL microtube to obtain a final reaction volume of 200 µL. Then, 1.25 µL of 10% Tween-20 (Millipore-Sigma) and 10 µL of a 400 mM solution of the reducing agent sodium cyanoborohydride (NaBH3CN; Millipore-Sigma, St. Louis, MO, USA) were added to the microtube. The tube was incubated for 18–24 h at 37 °C with mild agitation at 6–10 rpm. Thereafter, unreacted aldehyde groups on the beads were blocked by adding 10 µL of 65 mg/mL solution of carboxymethoxylamine (CMO, Millipore-Sigma) in 800 mM sodium hydroxide (NaOH; Millipore-Sigma) followed by a 60 min incubation at 37 °C under mild agitation. The uncoupled Erns protein was removed by centrifugation (16,000× *g* for 15 min at 4 °C). Then, it was washed two times with 200 µL of 100 mM Tris-HCl pH 8. The supernatant was then removed with a micropipette and the bead pellet resuspended in 200 µL of storage buffer (PBS pH 7.4 containing 0.05% Proclin-300 (Millipore-Sigma)). The bead solution was briefly sonicated with ultrasonic homogenizer (UX-050) (Mitsui-EC Corp, Tokyo, Japan) and then stored at 4 °C until assayed.

### 2.6. CSFV Erns IgG AlphaLISA Optimization

The CSFV Erns AlphaLISA was designed as an IgG assay (Figure 1). In brief, Erns IgG in the sample binds to Erns protein that conjugated to acceptor beads and biotinylated anti-swine IgG that conjugated to streptavidin donor beads. Binding of the Ern IgG and anti-swine IgG brings the donor and acceptor beads into proximity. Laser excitation (680 nm) of the donor beads releases singlet oxygen molecules that trigger a cascade of chemical events in acceptor beads and results in a chemiluminescent emission (615 nm) detected by the microplate reader.

To create the CSFV Erns AlphaLISA, sample dilution, incubation time, Erns protein-conjugated acceptor bead concentration, streptavidin donor bead concentration, and biotinylated goat anti-swine IgG concentration were systematically evaluated to optimize the signal and minimize background. For example, the optimal concentration of biotinylated goat anti-swine IgG was determined by evaluating concentrations of 10, 20, 40, 80, 100 nM for serum and concentrations of 20, 40, 80, 100, 160 nM for oral fluids (Figure 2).

The final one-hour procedure consisted of two steps. First, 5 µL of sample (serum or oral fluid) or control (negative and positive) was incubated for 30 min with 20 µL of a mix containing 25 µg per mL Erns protein-conjugated AlphaLISA acceptor beads and 80 nM biotinylated goat anti-swine IgG (Fc) (Bethyl Laboratories Inc., Montgomery, TX, USA) in assay buffer 1x (AlphaLISA buffer 10x; PerkinElmer, Waltham, MA, USA). In-house positive and negative controls (5 µL) were included on each plate (in duplicate) for both serum and oral fluid assays. In-house negative controls were serum samples obtained from unvaccinated and uninoculated pigs and positive controls were serum samples from pigs infected with CSFV. In the second step, 25 µL of 80 µg per mL streptavidin donor beads (PerkinElmer) was added to each well and the plates incubated for 30 min. Incubations were performed at room temperature (22–25 °C) in half-area 96-well white plates (PerkinElmer) while protected from light and held on a rotating shaker (200 rpm). Thereafter, plates were read (615 nm) using a microplate reader (Envision^®^ 2105 multimode plate reader, PerkinElmer, Waltham, MA, USA) operated with the Envision Manager software version 1.14 (PerkinElmer). For quality purposes, negative control AlphaLISA signals (counts) were required to range from 0–400 and positive controls from 4000–7000. Sample AlphaLISA responses were converted to sample-to-positive (S/P) ratios:(1)$$S/P\;\mathrm{ratio}=\frac{\mathrm{Sample}\;\mathrm{AlphaLISA}\;\mathrm{signal}\;-\;\mathrm{Negative}\;\mathrm{control}\;\mathrm{signal}\;\left(\mathrm{mean}\right)}{\mathrm{Positive}\;\mathrm{control}\;\mathrm{signal}\;\left(\mathrm{mean}\right)-\;\mathrm{Negative}\;\mathrm{control}\;\mathrm{signal}\;\left(\mathrm{mean}\right)}$$

### 2.7. Statistical Analysis

CSFV S/P values for serum and oral fluids were analyzed by ANOVA and post hoc Tukey’s pairwise comparisons. ROC analyses were done to evaluate the performance of the serum and oral fluid CSFV Erns AlphaLISAs using R software package pROC [27]. Diagnostic sensitivity and specificity were derived from the ROC analyses for specific assay cutoffs. In the ROC analyses, samples from Groups 1 and 4 were classified as Erns antibody-negative and samples from Groups 2 and 3 collected ≥ 14 DPI or DPV were considered positive. Associated 95% CIs for diagnostic sensitivity and specificity were estimated using a procedure for normally distributed correlated data [28]. In brief, the correlation of the data was taken account by fitting normalized data into a linear mixed model.
*Y_ij_* = *µ* + *γ_i_* + *τs_ij_* + *ϵ_ij_*(2)

In Equation (2), *Y_ij_* is the *j*^th^ observation for the *i*^th^ subject; *µ* is the overall mean for samples classified as CSFV antibody-negative; *γ_i_* is the random effect of the *i*^th^ subject; *τ* is the fixed effect indicating the mean difference between CSFV antibody-negative and positive groups; *s_ij_* is the disease status of the *j*^th^ observation for the *i*^th^ subject; and *ϵ_ij_* is the random error of the *j*^th^ observation for the *i*^th^ subject. The variances from Equation (2) were used to calculate the 95% CIs for ROC-derived diagnostic sensitivity and diagnostic specificity estimates. Logit transformation was used to prevent the estimated intervals from exceeding the range of probability, i.e., [0, 1]. Prior to performing the ROC analyses, the S/P data of serum and oral fluid AlphaLISAs were normalized by 1/5 and 1/4 power transformations, respectively.

## 3. Results

### 3.1. CSFV Antibody Responses

Serum and oral fluid CSFV VNT and Erns IgG AlphaLISA antibody responses are shown in Table 1 by group. All serum samples (n = 160) from Group 1 CSFV-naïve pigs were negative by VNT and all serum samples from Groups 2, 3, and 4 were VNT-positive (neutralizing antibody titers ≥ 2). Within Groups 2, 3, and 4, Tukey’s pairwise comparison tests detected no difference in S/P responses between serum and oral fluids. However, comparisons between Groups 2, 3, and 4 showed higher Erns S/P responses in both serum and oral fluids from Group 2 pigs versus those in serum and oral fluids from Groups 3 or 4.

### 3.2. Evaluation of Test Diagnostic Performance

The results of the ROC curve analysis of the CSFV Erns IgG AlphaLISA based on the aggregate testing data for serum and oral fluid are shown in Table 2. At a cutoff of S/P ≥ 0.7, the aggregate estimated diagnostic sensitivities and specificities of the assay were, respectively, 97.4% (95% CI 95.9, 98.3) and 100% for serum and 95.4% (95% CI 92.9, 97.0) and 100% for oral fluid.

The results of the ROC analysis by treatment group are given in Table 3. Using a cutoff of S/P ≥ 0.7, the diagnostic specificity was 100% for both serum and oral fluid samples from the negative control (Group 1) and DIVA-vaccinated animals (Group 4). In ALD wild-type-inoculated (Group 2) pigs, the diagnostic sensitivities for serum and oral fluids were 98.0% (95% CI 93.0, 99.8) and 92.9% (95% CI 88.2, 96.2), respectively. In LOM strain-vaccinated (Group 3) pigs, the diagnostic sensitivities for serum and oral fluid tests were 96.4% (95% CI 91.9, 98.3) and 96.7% (93.0, 98.8), respectively.

## 4. Discussion

Detectable levels of CSFV-specific antibody generally appear 10–15 days after infection or vaccination [25,29], with protective immunity conferred by the neutralizing antibody against E2 and Erns envelope glycoproteins [30]. CSFV neutralizing antibodies are measured by virus neutralization tests (VNT), which mainly detect E2-specific antibodies. The VNT is considered a confirmatory test for antibody detection and the most suitable method for discriminating CSFV-specific antibodies from antibodies against other pestiviruses that infect pigs, i.e., BVDV, BDV, APPV, and others [8]. In this study, a VNT was performed to provide information regarding the antibody status of infected or vaccinated pigs.

The results from the current study support previous reports that oral fluid CSFV antibody assays are technically possible and can provide similar diagnostic sensitivity and specificity to that achieved with serum [26]. Oral fluid samples from individual pigs or pens of pigs are easily collected and, for that reason, routinely used by pork producers and swine veterinarians for the detection of various swine diseases [31]. Oral fluid assays compatible with high-throughput testing, i.e., AlphaLISA, can enhance the effective CSFV surveillance approach at the population level.

Likewise, the ability to rapidly identify CSFV infections with an easy-to-use method would facilitate surveillance. That is, the CSFV Erns IgG AlphaLISA has a total turnaround time of one hour, with an identical protocol used for both serum and oral fluid specimens. Serum and oral fluid Erns AlphaLISA assays likewise provide flexibility for specific CSFV situations. That is, by using different cutoffs, the rate of serum and oral fluid false negative and false positive results from CSFV Erns antibody-negative (Groups 1 and 4) and -positive (Groups 2 and 3) groups could be adjusted for the immediate surveillance objective (Table 2). Higher S/P cutoffs (e.g., S/P ≥ 0.7) could be used to increase the test’s diagnostic specificity when screening presumed negative populations. In contrast, the use of lower S/P cutoffs could be used to increase its diagnostic sensitivity, i.e., when in a “case-finding” mode after a confirmed case.

The Erns IgG AlphaLISA developed in this study provided high diagnostic sensitivity and specificity for detecting CSFV Erns antibody in serum. At a cutoff of 0.7, the assay showed a diagnostic sensitivity of 97.4% with serum from CSFV-infected pigs, and a diagnostic specificity of 100% with sera from pigs vaccinated once with CP7_E2alf. In contrast, a previous report evaluating commercial Erns antibody ELISAs from various laboratories and sample sets showed a diagnostic sensitivity of 90–98% with sera from CSFV-infected pigs and a diagnostic specificity of 89–96% with sera from CP7_E2alf-vaccinated pigs [20]. For oral fluids and using the same cutoff (0.7), the Ern IgG AlphaLISA showed a diagnostic sensitivity of 92.9% with individual oral fluids from CSFV-infected pigs and a diagnostic specificity of 100% with pen-based oral fluids from pigs vaccinated once with CP7_E2alf. There is only one prior report of a CSFV oral fluid antibody assay, i.e., an IgG Erns ELISA [27]. In that study, an IgG Erns ELISA provided a diagnostic sensitivity of 97.5% and diagnostic specificity of 100% for individual pig oral fluid samples from CSFV-infected or live virus-vaccinated pigs, and from uninfected and unvaccinated pigs, respectively [27]. However, the specificity of the oral fluid Erns ELISA was not evaluated for the DIVA properties in that study.

The AlphaLISA described in this study was specifically designed to detect CSFV-specific Erns antibodies, the basis of CSFV DIVA testing. The assay was evaluated using samples from pigs vaccinated with the marker vaccine based on a chimeric pestivirus (CP7_E2alf). The vaccine is based on the BVDV-1 strain CP7 in which the E2 protein coding sequence was replaced by the corresponding sequence of the CSFV strain Alfort/187 [32]. Thus, Erns-specific IgG antibodies were detected in serum and oral fluid samples from pigs inoculated with a wild-type CSFV isolate (Group 2) or from pigs vaccinated with a modified live vaccine (Group 3), but not in pigs vaccinated with the DIVA vaccine (Group 4). In light of the strong data presented here and given the significant animal health implications of these results, additional studies evaluating the performance of the assay are justified, e.g., repeatability, reproducibility, cross-reactivity with other pestiviruses, as well as responses in DIVA-vaccinated and -challenged pigs.

## 5. Conclusions

We developed and optimized an AlphaLISA system for detecting serum and oral fluid CSFV Erns IgG antibody in swine. The assay provided excellent diagnostic performance in terms of detecting Erns antibody in CSFV-infected or whole virus-vaccinated pigs and demonstrated the ability to differentiate pigs vaccinated with CSF E2-based DIVA vaccines.

## Figures and Tables

**Figure 1 animals-13-03802-f001:**
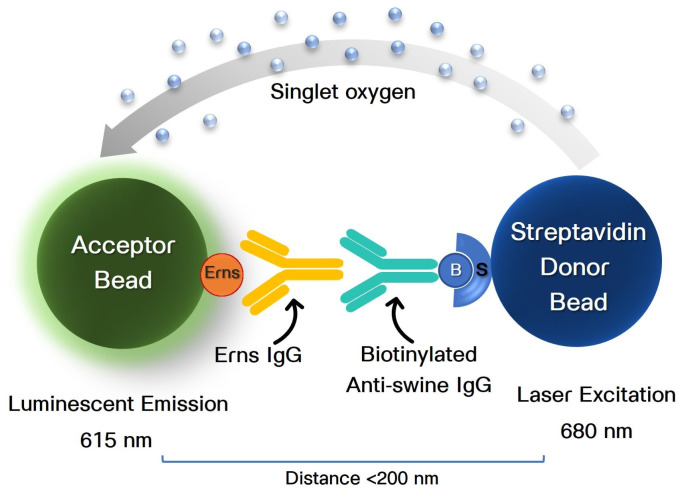
Classical swine fever virus (CSFV) amplified luminescent proximity homogenous assay (AlphaLISA^®^) platform for detection of CSFV Erns IgG antibody. In the presence of Erns IgG in a tested sample (serum, oral fluid), the Erns IgG binds Erns protein that conjugated to acceptor beads, and the biotinylated anti-swine IgG that conjugated to streptavidin donor beads. Binding of the antibody brings the donor and acceptor beads into proximity. Excitation of the donor beads at 680 nm promotes the release of singlet oxygen molecules that trigger a cascade of chemical events in the acceptor beads, resulting in a light emission at 615 nm.

**Figure 2 animals-13-03802-f002:**
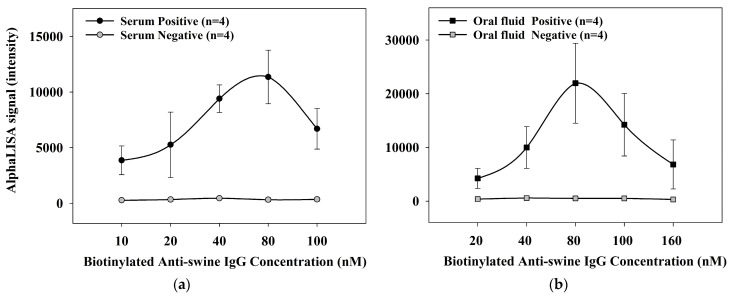
Optimization of biotinylated anti-swine IgG antibody concentrations for serum and oral fluid AlphaLISA^®^. The optimum combination of a mix containing 25 µg/mL of Erns protein-conjugated acceptor beads and 80 µg/mL of streptavidin donor beads were tested with various biotinylated anti-swine IgG concentrations: (**a**) negative and positive sera were evaluated at 10, 20, 40, 80, and 100 nM; (**b**) negative and positive oral fluids were tested at 20, 40, 80, 100, and 160 nM. The AlphaLISA^®^ signal represents the mean values (n = 4) with error bars denoting the standard deviation.

**Table 1 animals-13-03802-t001:** Classical swine fever virus (CSFV) antibody responses by group and specimen.

Group and CSFV Status	Specimen (No. Samples)	Virus Neutralization Geometric Mean Titer (log_2_ x) (95% CI)	Erns IgG AlphaLISA Mean Sample-to-Positive (S/P) Ratio (95% CI)
1. Negative (unvaccinated and uninoculated).	Serum (n = 160)	negative	−0.01 ^a^ (−0.01, 0.00)
	OF from individual pigs (n = 160)	•	−0.00 ^a^ (−0.00, 0.00)
2. Inoculated CSFV ALD wild-type strain.	Serum (n = 100)	6.17 (5.82, 6.52)	7.48 ^b^ (6.55, 8.40)
	OF from individual pigs (n = 184)	•	7.29 ^b^ (5.90, 8.68)
3. Vaccinated CSFV LOM live vaccine.	Serum (n = 100)	4.01 (3.79, 4.23)	1.93 ^c^ (1.76, 2.11)
	OF from individual pigs (n = 184)	•	1.89 ^c^ (1.80, 1.99)
4. Vaccinated CSF marker live vaccine.	Serum (n = 360)	5.09 (4.91, 5.27)	0.05 ^a^ (0.04, 0.06)
	OF from pens of pigs (n = 30)	•	0.01 ^a^ (−0.01, 0.04)

• = not tested; OF = oral fluids; ^a^–^c^ = different superscripts indicate that the mean of S/P responses differed significantly between treatment groups or specimen types (*p* < 0.05).

**Table 2 animals-13-03802-t002:** Classical swine fever virus (CSFV) Erns IgG AlphaLISA aggregate (Groups 1, 2, 3, and 4 ^‡^) diagnostic sensitivity (Dx Se) and diagnostic specificity (Dx Sp) by cutoff.

Cutoff (S/P)	Serum		Oral Fluids	
	Dx Se (95% CI)	Dx Sp (95% CI)	Dx Se (95% CI)	Dx Sp (95% CI)
≥0.1	100 (100, 100) *	88.3 (85.1, 90.8)	99.7 (99.2, 99.9)	100 (100, 100) *
≥0.2	100 (100, 100) *	94.8 (93.0, 96.2)	99.7 (99.4, 99.9)	100 (100, 100) *
≥0.3	97.9 (96.3, 98.8)	99.8 (98.6, 99.9)	99.7 (99.5, 99.9)	100 (100, 100) *
≥0.4	97.9 (96.5, 98.8)	99.8 (99.7, 99.9)	99.7 (99.5, 99.8)	100 (100, 100) *
≥0.5	97.4 (95.7, 98.4)	99.8 (99.7, 99.9)	99.5 (99.1, 99.7)	100 (100, 100) *
≥0.6	97.4 (95.8, 98.4)	99.8 (99.7, 99.9)	97.0 (95.2, 98.1)	100 (100, 100) *
≥0.7	97.4 (95.9, 98.3)	100 (100, 100) *	95.4 (92.9, 97.0)	100 (100, 100) *
≥0.8	97.4 (96.0, 98.3)	100 (100, 100) *	93.5 (90.2, 95.7)	100 (100, 100) *
≥0.9	97.4 (96.0, 98.3)	100 (100, 100) *	92.1 (88.5, 94.7)	100 (100, 100) *
≥1.0	97.4 (96.1, 98.3)	100 (100, 100) *	91.3 (87.5, 94.0)	100 (100, 100) *

* 95% CIs were estimated at sensitivity/specificity = 99.99% because logit (100%) is undefined. ^‡^ Group 1, negative controls; Group 2, inoculated with ALD strain wild-type CSFV; Group 3, vaccinated with LOM strain live CSFV vaccine; Group 4, vaccinated with CSF marker vaccine.

**Table 3 animals-13-03802-t003:** Classical swine fever virus (CSFV) antibody positivity (%, 95% CI) by group, specimen type, and assay.

Group and CSFV Status	Specimen (No. Samples)	VNT	Erns IgG AlphaLISA by Cutoff		
			S/P ≥ 0.3	S/P ≥ 0.5	S/P ≥ 0.7
1. Negative (unvaccinated and uninoculated).	Serum (n = 160)	0.0 (0.0, 2.3)	0.0 (0.0, 2.3)	0.0 (0.0, 2.3)	0.0 (0.0, 2.3)
	OF from individual pigs (n = 160)	•	0.0 (0.0, 2.3)	0.0 (0.0, 2.3)	0.0 (0.0, 2.3)
2. Inoculated CSFV ALD wild-type strain.	Serum (n = 100)	100 (96.4, 100)	98.0 (93.0, 99.8)	98.0 (93.0, 99.8)	98.0 (93.0, 99.8)
	OF from individual pigs (n = 184)	•	100 (98.0, 100)	100 (98.0, 100)	92.9 (88.2, 96.2)
3. Vaccinated CSFV LOM live vaccine.	Serum (n = 100)	100 (97.4, 100)	96.4 (91.9, 98.3)	96.4 (91.9, 98.3)	96.4 (91.9, 98.3)
	OF from individual pigs (n = 184)	•	99.5 (97.0, 100)	99.5 (97.0, 100)	96.7 (93.0, 98.8)
4. Vaccinated CSF marker live vaccine.	Serum (n = 360)	100 (99.0, 100)	1.1 (0.3, 2.2)	0.3 (0.0, 1.5)	0.0 (0.0, 1.0)
	OF from pens of pigs (n = 30)	•	0.0 (0.0, 11.6)	0.0 (0.0, 11.6)	0.0 (0.0, 11.6)

• = not tested; OF = oral fluids; VNT = virus neutralization test; S/P = sample-to-positive ratio.

## Data Availability

All data generated or analyzed during this study are included in this published article.

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
