# Peer review of "Performance of a Differentiation of Infected from Vaccinated Animals (DIVA) Classical Swine Fever Virus (CSFV) Serum and Oral Fluid Erns Antibody AlphaLISA Assay"

_animals, 2023, doi:10.3390/ani13243802_

Round 1
Reviewer 1 Report
Comments and Suggestions for Authors
In this manuscript titled “Performance of a DIVA Classical Swine Fever Virus (CSFV) Serum and Oral fluid Erns Antibody AlphaLISA assay”, the DIVA Classical Swine Fever Virus (CSFV) Serum and Oral fluid Erns Antibody AlphaLISA assay was performed. The combined use of a marker vaccine to protect against clinical disease and a serum/oral fluids DIVA assay able to identify wild- type CSFV infections will greatly improve the capacity of the pork industry to deal quickly and decisively with CSFV. The manuscript is well-organized and has certain significance. It was addressed a specific gap in the field. However, there are still some problems in this manuscript that need to be revised;
The Erns glycoprotein is absent in the CSFV marker vaccine, hence vaccinated pigs are
negative on the CSFV Erns AlphaLISA®. Importantly for ease of use in the field and in the labor-
atory, the assay is able to test both serum and oral fluid samples. The combined use of a marker vaccine to protect against clinical disease and a serum/oral fluids DIVA assay able to identify wild-type CSFV infections.
The combined use of a marker vaccine to protect against clinical disease and a serum/oral fluids DIVA assay able to identify wild- type CSFV infections will greatly improve the capacity of the pork industry to deal quickly and decisively with CSFV. The Erns IgG antibody AlphaLISA combined DIVA capa- bility with solid diagnostic performance, rapid turnaround, ease of use, and compatibility with both serum and oral fluid specimens.
1 The language needs considerable attention.
2 The picture quality is generally poor(figure1 and 2, 3).I cannot see clearly.
Author Response
Thank you for your comments. All comments have contributed to the improvement of the manuscript. Our responses to the specific comments are given in an attached file and highlighted in yellow in the revised manuscript

Reviewer 2 Report
Comments and Suggestions for Authors
The authors of the present manuscript have optimized the AlphaLISA assay for the classical Swine Fever Virus (CSFV) to function as a tool for differentiating between infected and vaccinated animals (DIVA). CSFV poses a significant threat to the swine industry as one of the major viral diseases affecting swine. The creation of a DIVA marker vaccine and a corresponding detection assay is crucial in differentiating naturally infected animals from those that have been vaccinated. This differentiation aids in controlling the spread of the disease. The primary focus of the current manuscript is on the development of the DIVA marker assay, aiming to contribute to the effective management of CSFV transmission.
However, a few minor comments on the manuscript are noted below:
1. Line 64-65: Kindly provide a reference for the information presented in this sentence.
2. Could the authors specify the infectious doses utilized for the inoculation/immunization of pigs with CSFV strains in this study?
3. It would be valuable for the authors to present the results of the AlphaLISA assay based on Days Post Vaccination (DPV) or Days Post Infection (DPI) rather than combining them into a single dataset. This approach could enhance the understanding of the assay's sensitivity and specificity in differentiating infected animals at various stages of infection.
4. There appears to be a discrepancy in the sample size for Group 2 serum in Tables 1 and 3. While the tables indicate a sample size of 100, the materials and methods section depicts a sample size of 140. Correct this inconsistency.
5. The strain of CSFV chosen for the production of the Erns protein is not mentioned in the provided text.
Author Response
Thank you for your comments. All comments have contributed to the improvement of the manuscript. Our responses to specific comments are given (please see the attachment), and highlighted in yellow in the revise manuscript.

Reviewer 3 Report
Comments and Suggestions for Authors
The authors report the development and validation of a novel antibody test for the detection of antibodies (IgG) specific to the classical swine fever virus (CSFV) antigen Erns. The assay is based on the AlphaLISA system. The purpose of the assay would be to support the application of differentiating infected from vaccinated animals (DIVA) using antibody responses to the Erns antigen. Over all the study is interesting and the assay reported has the potential to improve the application of DIVA principles in improving the control of CSFV. As a diagnostic platform the AlphaLISA system offers excellent sensitivity and specificity with reduced testing time required. The manuscript will be of interest to those working on this important virus.
Line 51 to 55 – Suggest breaking this very long sentence into multiple sentences.
Line 56 suggest revision “Differentiating infected from vaccinated animals (DIVA) enabling vaccines and companion”
Line 70-72 – I would suggest the authors expand this text to specifically describe how these previous studies have suggested further improvements are required.
Line 134 The text suggests the Erns Antigen produced as described in Ref 26. While it is described as “In brief”, it seems to be a very comprehensive description.
Lines 150 to 160, including Figure 1, these are technically results.
Is there an explanation for the multiple bands observed in the eluted Erns (Fig 1a) and final Erns sample (Fig 1b)?
What was the outcome of the Western blot experiment (line 149)?
As mentioned above these are technically results, but in the context of what they show, Figure 1 may be better provided as a supplemental figure.
Line 234 Should be labelled as Equation 1
Line 248 Should be labelled as Equation 2
Line 269 Table 2 (and Table 3) the abbreviation “OF” should be explained in either table title or as a footnote.
Comments on the Quality of English LanguageThere are some long sentences in the manuscript that could be revised into multiple sentences to improve the overall readability of the manuscript.
Author Response
Thank you for your comments. All comments have contributed to the improvement of the manuscript. Our responses to specific comments are given (please see the attachment) and highlighted in yellow in the revised manuscript.
